# Direct observation of nanoscale dynamics of ferroelectric degradation

Qianwei Huang[1], Zibin Chen [1✉], Matthew J. Cabral[1], Feifei Wang[2], Shujun Zhang [3], Fei Li [4], Yulan Li[5], Simon P. Ringer [1], Haosu Luo [6], Yiu-Wing Mai[1] & Xiaozhou Liao [1✉]

Failure of polarization reversal, i.e., ferroelectric degradation, induced by cyclic electric loadings in ferroelectric materials, has been a long-standing challenge that negatively impacts the application of ferroelectrics in devices where reliability is critical. It is generally believed that space charges or injected charges dominate the ferroelectric degradation. However, the physics behind the phenomenon remains unclear. Here, using in-situ biasing transmission electron microscopy, we discover change of charge distribution in thin ferroelectrics during cyclic electric loadings. Charge accumulation at domain walls is the main reason of the formation of *c* domains, which are less responsive to the applied electric field. The rapid growth of the frozen *c* domains leads to the ferroelectric degradation. This finding gives insights into the nature of ferroelectric degradation in nanodevices, and reveals the role of the injected charges in polarization reversal.

[1] School of Aerospace, Mechanical and Mechatronic Engineering, The University of Sydney, Sydney, NSW, Australia. [2] Key Laboratory of Optoelectronic Material and Device, Department of Physics, Shanghai Normal University, Shanghai, China. [3] Institute for Superconducting and Electronic Materials, Australian Institute of Innovative Materials, University of Wollongong, Wollongong, NSW, Australia. [4] Electronic Materials Research Laboratory, Key Laboratory of the Ministry of Education, Xi'an Jiaotong University, Xi'an, China. [5] Pacific Northwest National Laboratory, Richland, WA, USA. [6] Key Laboratory of Inorganic Functional Materials and Devices, Shanghai Institute of Ceramics, Chinese Academy of Sciences, Shanghai, China. ✉email: z.chen@sydney.edu.au; xiaozhou.liao@sydney.edu.au

Ferroelectric materials have been extensively used in electronic nanodevices because of their polarization reversal behavior under an applied electric field[1–4]. However, the application of repeated electric fields has been shown to reduce the ferroelectric performance, leading to polarization reversal failure, i.e., ferroelectric degradation[5]. Ferroelectric degradation results in short life cycles, which impacts the reliability and endurance of electronic nanodevices[6]. Ferroelectric degradation changes the physical properties of ferroelectrics, including coercive field[7], remnant polarization[8,9], and resistivity[10] under cyclic electric loadings. With the increasing usage of nanoscale electronic devices, a comprehensive understanding of the mechanisms and evolution of ferroelectric degradation is essential.

Ferroelectric degradation is often linked to numerous mechanisms, including domain wall pinning, domain nucleation suppression, and microcracks[6,7,11]. The domain wall pinning mechanism claims that the charge carriers and/or defect dipoles accumulate at the domain walls during electric cycling, leading to domain locking[12–14]. Some studies proposed a competitive process between domain wall pinning and unpinning, where the detrapping of charge carriers also occurred during ferroelectric degradation to unpin the domain walls. Ferroelectric degradation occurs once the pinning rate dominates[15,16]. The domain nucleation suppression mechanism suggests that charge carriers injected from the electrode prohibit the nucleation and growth of domains near the sample surface, degrading the ferroelectric performance[17]. The microcrack mechanism states that long-term electric cycling could cause the formation of microcracks initiated from the grain boundaries or domain walls[7,18,19].

Ferroelectric degradation is usually attributed to the activity of excess charges from the space or injection[7,20,21], which is generally accumulated at the domain wall under cyclic electric field and accounts for the degraded polarization[22]. However, it is not clear how these charges lead to the ferroelectric degradation. Direct observation of charge accumulation and real-time evolution of domain structure with cyclic electric field is critical to understand the impact of charge activities on ferroelectric degradation[7,17]. Owing to the advancements of in-situ transmission electron microscopy (TEM), real-time nanoscale observation of ferroelectric domain switching behavior under electric fields has become possible[23–25]. In addition, the recently developed microelectromechanical system (MEMS)-based nano-chip technique for TEM[26] have improved the stability and reliability of the experiment under repeated electric fields, and therefore experiments revealing the mechanism of ferroelectric degradation have become technically feasible.

In this study, the domain switching behavior of Pb(Mg$_{1/3}$Nb$_{2/3}$)O$_3$–0.38PbTiO$_3$ (PMN-0.38PT) single crystals was investigated using in-situ biasing TEM with MEMS-based nano-chips, where an in-plane electric field was repeatedly applied to the sample. Scanning transmission electron microscopy-differential phase contrast (STEM-DPC) imaging was used to explore the charge distribution at domain walls during cyclic electric loading. With the increase of the electric loading cycles, domains with out-of-plane polarization (c domains) were initiated from the original domain walls, during which the charge accumulation was responsible for the nucleation and growth of the c domains. Of particular interest is that the c domains were less responsive to the applied electric field, showing a frozen state under electric field, which is a critical factor in ferroelectric degradation. The discovery of this ferroelectric degradation mechanism not only strengthens the understanding of ferroelectric degradation at nanoscale, but also provides guidance on the design of ferroelectric nanodevices.

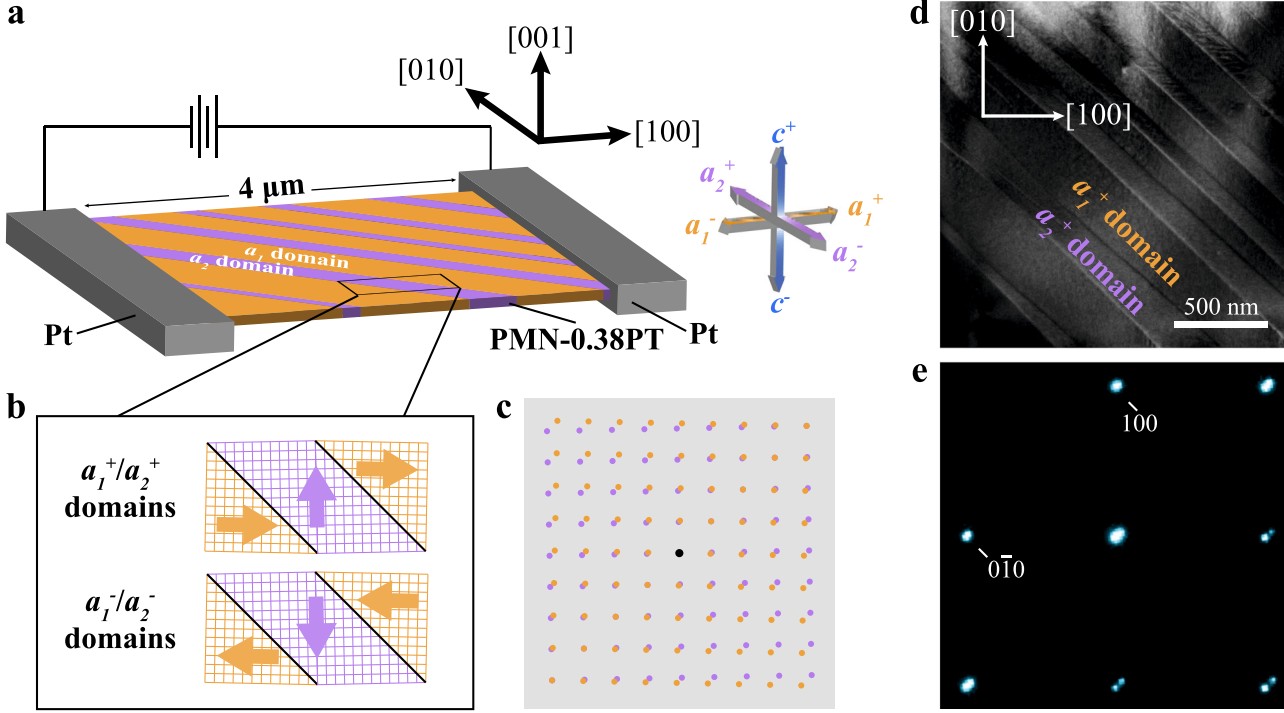

**Fig. 1 In-situ biasing TEM setup and determination of initial $a_1/a_2$ 90° domains. a** A schematic diagram of the experimental setup. A focused ion-beam fabricated PMN-0.38PT lamella was fixed between two Pt electrodes. A cyclic electric field was applied to the lamella along ±[100] crystallographic direction of the lamella. The yellow, purple, and blue arrows denote the polarizations of $a_1$ ($a_1^+/a_1^-$), $a_2$ ($a_2^+/a_2^-$), and c ($c^+/c^-$) domains, which are along ±[100], ±[010], and ±[001] orientations, respectively. **b** Schematic diagrams showing [001] projected lattices of $a_1/a_2$ domains, and **c** their corresponding schematic of electron diffraction pattern showing the split spots. **d** A TEM image showing 90° stripe domains in PMN-0.38PT, and **e** the corresponding electron diffraction pattern showing the split spots of the 90° domains.

## Results

**Determination of initial $a_1/a_2$ 90° domains**. Figure 1 shows a schematic of the in-situ TEM experiment setup (Fig. 1a) and the initial domain structure of a PMN-0.38PT single crystal before cyclic electric loading (Fig. 1b–e). As PMN-0.38PT is a ferroelectric material with a tetragonal structure at room temperature[27], there are six possible polarization directions in PMN-0.38PT along $\pm[100]$, $\pm[010]$, and $\pm[001]$ orientations, which are defined as $a_1$ ($a_1^+/a_1^-$), $a_2$ ($a_2^+/a_2^-$), and $c$ ($c^+/c^-$) domains, respectively, as indicated in Fig. 1a. A PMN-0.38PT lamella with stripe domains prepared by focused ion-beam was fixed between the two Pt electrodes with a distance of 4 μm apart. A cyclic electric field was applied to the lamella along the $\pm[100]$ crystallographic direction. The stripe domains with domain walls along the $[1\bar{1}0]$ direction indicate that they are $a_1$ and $a_2$ domains with 90° ferroelastic domain walls[28]. To minimize the system energy, the 90° stripe domains follow the head-to-tail configuration, in which the polarizations of the domains in Fig. 1a are either $a_1^+/a_2^+$ ($\rightarrow/\uparrow$) or $a_1^-/a_2^-$ ($\leftarrow/\downarrow$) (Fig. 1b). Figure 1b, c presents schematic diagrams of [001] projected lattices of $a_1/a_2$ domains and the corresponding schematic of electron diffraction pattern, respectively. The electron diffraction pattern shows the spot splitting caused by the 90° polarization rotation of $a_1$ domains and $a_2$ domains. The yellow and purple spots are contributed by the yellow ($a_1$ domains) and purple ($a_2$ domains) lattices, respectively. Figure 1d, e shows TEM images of stripe domains in the lamella and the corresponding electron diffraction pattern of the domains, respectively. Under an applied electric field along the $\pm[100]$ direction, the polarization of $a_1$ domains will align with the direction of the electric field[29]. Based on this switching behavior of $a_1$ domains under positive and negative electric fields, the pair of the domains was determined to be $a_1^+/a_2^+$.

**Formation of $c$ domains under cyclic electric field**. Figure 2 shows the evolution of domain structure in PMN-0.38PT under cyclic electric loading. The parameter set for the repeated electric biasing test was a bipolar triangle wave with a peak voltage of ±4.7 V and a frequency of 1/30 Hz. The electric field $E$ applied to the lamella can be calculated using the equation: $E = \frac{V}{d}$, where $V$ and $d$ are the applied voltage and the distance between the electrodes, respectively. Since the electric field between the two electrodes is homogeneously distributed[30] and the distance between the two electrodes is 4 μm, the peak applied electric field is ±1.175 MV/m. The pristine domain structure is presented in Fig. 2a (0 cycle), with $a_1^+/a_2^+$ domain configuration before the application of a cyclic electric field. After 130 cycles, some domains nucleated from the original domain walls (blue domains in the schematic). As the number of electric cycles was increased, additional domains nucleated from regions of the domain wall while earlier formed domains propagated along the [010] direction in the $a_1$ domains and the [100] direction in the $a_2$ domains. Electron diffraction was used to determine the polarization direction of these domains. Figure 2b, c presents electron diffraction patterns obtained from areas marked with the green and red circles in Fig. 2a, respectively. The diffraction spots acquired from the selected area of the green circle split along the $[100]^*$ direction (Fig. 2b), and those acquired from the red circle split along the $[010]^*$ direction (Fig. 2c). The lattice parameter of PMN-0.38PT along the [100] direction is a little larger than that along the [010] direction for the $a_1^+$ domains, while for $c$ domains, the lattice parameters along the [100] and [010] are equal. This leads to the split diffraction spots along the $[100]^*$ direction (Fig. 2d), which is consistent with the highlight in the green square of Fig. 2b, implying the coexistence of $a_1^+$ and $c$ domains in the green circle in Fig. 2a. Meanwhile, the coexistence

of $a_2^+/c$ domains also follows a similar scenario (Fig. 2e), represented by the highlight in the red square of Fig. 2c. Figure 2f shows the curve of the area of the projected $c$ domains vs. the number of cycles. The growth rate of the $c$ domains increased exponentially with the number of electric cycles.

It should be noted that nanodomains formed in some of the original $a_2$ domains during electric cycling and extended along the $[1\bar{1}0]$ direction (denoted by the red arrow in Fig. 2a). Literature reports suggest that mechanical stress could introduce these kinds of nanodomains in PMN-0.38PT[28,29]. In the current research, the two ends of the lamella were fixed onto the electrodes, which constrained the deformation from domain switching and induced the local stress that was the driving force for the formation of the nanodomains.

**Charge accumulation at domain walls under cyclic electric loadings**. It has been reported that ferroelectric domain walls are preferred regions for the accumulation of oxygen vacancies and other structural defects[31,32]. With increasing the number of cyclic electric loadings, an increasing number of mobile charges will accumulate at domain walls due to the electrostatic effect[12]. The accumulation of the space charges or injected charges changes the local electric field in the ferroelectric materials[33]. The STEM-DPC technique can image local electric field[34,35] with high accuracy, which makes it possible to monitor the evolution of local electric fields in ferroelectrics under external stimuli. Our experimental results suggest that the repeated electric loading processes did not change the focus and tilt (Supplementary Fig. 1) and that defocus condition did not change the electron field maps observed from STEM-DPC images within a certain defocus range (Supplementary Fig. 2). Figure 3a, b shows the local electric field maps, which were acquired using the STEM-DPC technique, of an area in a sample before and after the application of a cyclic electric field. The direction and amplitude of local electric fields are presented by colors, see the color wheel at the lower-left corner of Fig. 3a, and image intensity, respectively. After cyclic electric loadings, the direction of local electric fields (the colors in the image) remained the same. However, the amplitude of local electric fields (or local image intensity) at areas denoted by the white hollow arrows (mainly domain walls) increased. Figure 3c presents the result of Fig. 3a subtracted from Fig. 3b in the grayscale to show clearly the change of local image intensity. Significant contrast variation in areas marked by the white hollow arrows indicates the amplitude variation of local electric field, which was caused by the difference in local charge density before and after the application of cyclic electric field (the electron beam was blocked during the application of electric field to avoid any potential effect from electron beam illumination). Because there are few free charges before the application of electric field for as-grown PMN-0.38PT, the change of local charge density can only be an increase after the application of repeated electric field. Hence, the contrast variation in Fig. 3c is a result of charge accumulation. TEM images of the same area captured before (Fig. 3d) and after (Fig. 3e) cyclic electric loadings show no apparent difference, indicating that the change in local charge density did not change the local structure. To further validate the stability and reliability of STEM-DPC imaging, we provide additional data in Supplementary Figs. 3 and 4. With extra data from electron energy loss spectroscopy (EELS), a powerful technique to evaluate the electronic state of materials[36], we confirm the charge accumulation at domain walls. EELS was used to measure the plasma peak shift[37] at domain walls before and after cyclic electric loadings (Supplementary Fig. 5). The energy shift is 1.2 eV, corresponding to the change in charge density of 3.76 electrons/nm$^3$ at the domain walls (see the calculation in Supplementary note 1).

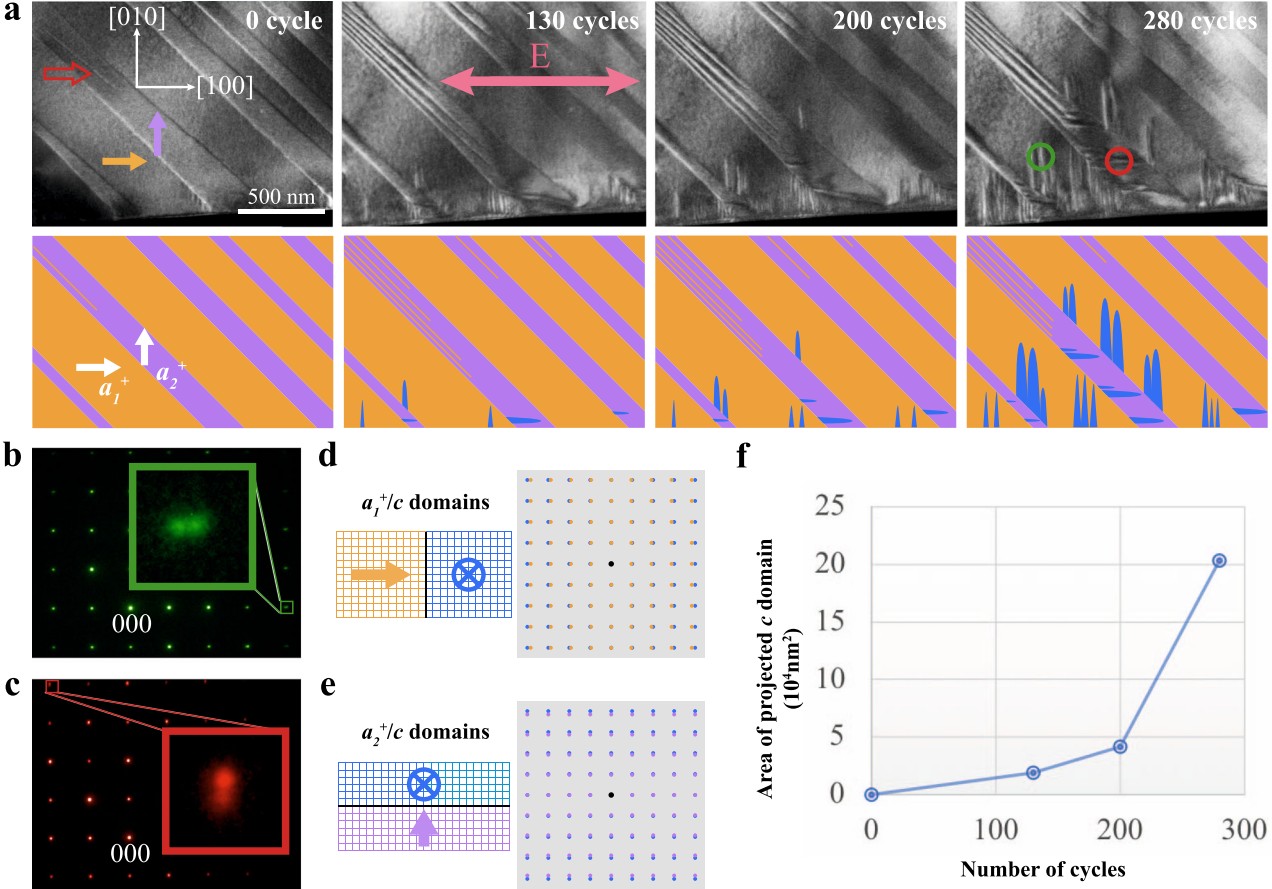

**Fig. 2 *c* domain formation during cyclic electric loading. a** A series of TEM images showing the evolution of domain structure in PMN-0.38PT under a cyclic electric field. The schematics of the domains were presented at the bottom of each image. The pink double arrow denotes the directions of the applied electric field. The red arrow denotes nanodomains formed in an original $a_2^+$ domain. **b, c** Electron diffraction patterns obtained from the areas marked with green and red circles in **a** after 280 electric cycles, respectively. The green and red squares denote the splitting of diffraction spots along the [100]* and [010]* directions, respectively (*represents reciprocal space). **d, e** Schematic diagrams of [001] projected lattices of $a_1^+/c$ and $a_2^+/c$ domains and their corresponding schematics of electron diffraction patterns. **f** The area of projected *c* domain vs. the number of cycles showing the rate of the *c* domain growth in **a** from 0 to 280 cycles.

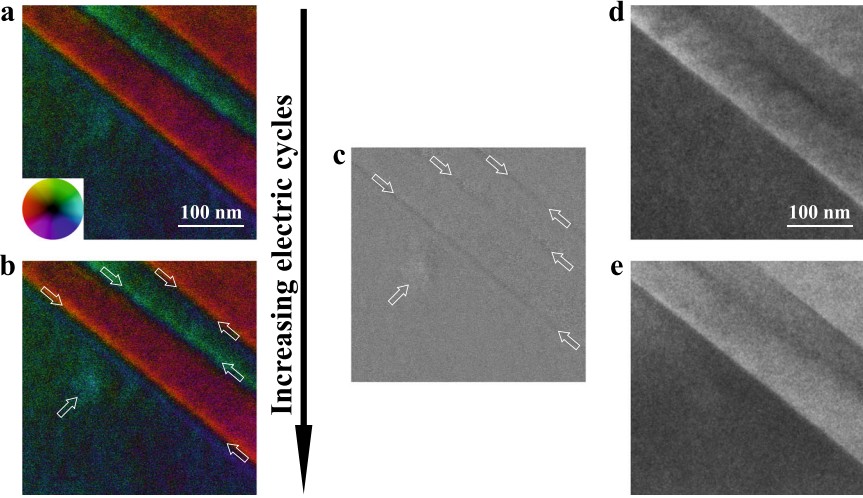

**Fig. 3 Charge accumulation at domain walls during cyclic electric loadings. a, b** Electric field mapping before and after cyclic electric loadings, respectively. The direction and amplitude of local electric fields are represented by colors, see the color wheel at the lower-left corner of **a**, and image intensity, respectively. **c** A grayscale image obtained from the subtraction of the image in **a** from the image in **b**, showing the variation in local image intensity. The white hollow arrows in **b** and **c** mark the places with significant intensity variation after cyclic loading. **d, e** TEM images before and after cyclic loadings, respectively.

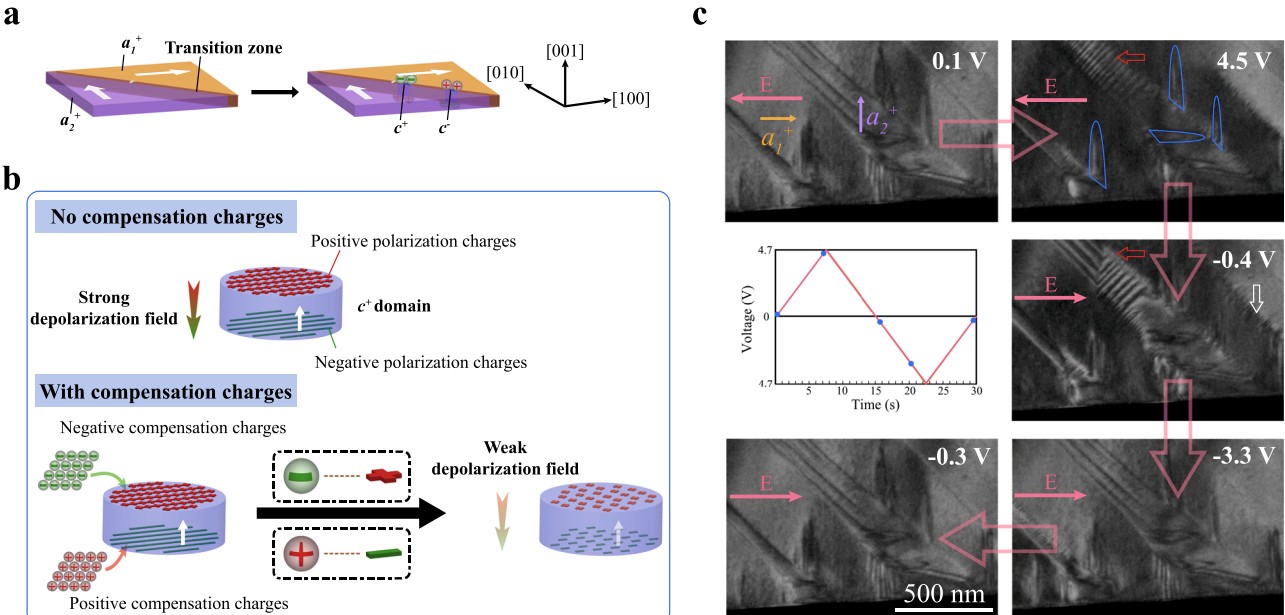

**Fig. 4 Stabilization of *c* domains by compensation charges and the frozen behavior of *c* domains under electric loading. a** The nucleation of *c* domains (the blue nuclei in the transition zone) from an original $a_1^+/a_2^+$ domain wall. **b** *c* domains were stabilized by compensation charges on the surfaces. **c** A voltage–time curve of one electric cycle and a series of TEM images (extracted from Supplementary Movie 1) showing the responses of *c* domains to the electric field stimulation. The *c* domains, which formed as a consequence of ferroelectric degradation, did not reverse with the application of the cyclic in-plane electric loading. The small red hollow arrows denote the intermediate ferroelastic domains formed during multi-switching in an $a_2^+$ domain. The small white hollow arrow denotes the zipper-shaped domain wall formed during the switching process. The blue lines mark the contours of some *c* domains.

**Stabilization of *c* domains by compensation charges**. A series of images in Fig. 2a show that the nucleation of the *c* domains occurred at the original 90° ferroelastic domain walls. Ferroelastic domain walls are polarization transition zones with a thickness of several unit cells[38]. Within the transition zones, the polarization rotates from $a_1^+$ to $a_2^+$, or from $a_2^+$ to $a_1^+$, minimizing the lattice mismatch between neighboring $a_1^+$ and $a_2^+$ domains[39]. Generally, it is difficult for polarization to rotate from in-plane to out-of-plane under in-plane applied electric field. However, the increase of accumulated mobile charges at the domain walls during the electric cycles cannot be accommodated by the in-plane $a_1$ and $a_2$ dipoles. Thus, metastable polarization in the transition zone is forced to switch to a direction that can accommodate these charges; and in this case, it is the ±[001] direction as shown in Fig. 4a[40]. Take a $c^+$ domain for example, after the polarization switching to the [001] direction, there are negative polarization charges at the lower surface and positive polarization charges at the upper surface as shown in Fig. 4b. The negative polarization charges attract positive accumulated charges while the positive polarization charges attract negative accumulated charges. These negative and positive accumulated charges rearrange at the specimen surface, serving as compensation charges that weaken the strong depolarization field and stabilize the *c* domains. Charge accumulation at domain walls is a dynamic process, while domain switching can only be accomplished when the charges accumulate to a certain level. That is, the formation of the *c* domains is not a result of a single electric loading, but a consequence of the cyclic electric loading. It is also indicated by the calculations that the electric field induced by the accumulated charges is strong enough to stabilize the *c* domains (see Supplementary note 1).

As the cyclic electric loading continues, the carrier concentration was also increased which drove the growth of the *c* domains (Fig. 2a). It was clearly observed in the TEM image series that *c* domains grew along the [010] direction in the $a_1^+$ domains and

the [100] direction in the $a_2^+$ domains. In addition, the domain walls that separate the $a_1^+/a_2^+$ and *c* domains are straight, which are actually the 90° ferroelastic domain walls, with the domains on both sides of the domain walls following a head-to-tail polarization configuration to minimize the system free energy[29].

**Frozen behavior of *c* domains under cyclic electric loading**. Thermodynamic calculations revealed that[41] the compensation charges not only weaken the depolarization field induced by polarization charges, but also strongly reduce the impact of external electric field. This indicates the *c* domains in our experiment may be less responsive to the in-plane electric field, being responsible for the observed ferroelectric degradation phenomena. Figure 4c presents how the *c* domains reacted to the applied electric field in one cycle (see Supplementary Movie 1). All the domains kept their initial configuration at a low bias field (0.1 V). As the positive bias gradually increased, the polarization of the original $a_1^+$ domains switched 180°. It is interesting to note that the *c* domains in the $a_1^+$ domains did not respond to the positive electric field and there was no change in the area of the *c* domains (the contours of some *c* domains are marked by blue lines in Fig. 4c). When a negative bias was applied, all domains remained the same (−0.4 V) until the bias reached the switching voltage. With the increasing magnitude of the negative electric field, all $a_1^+$ domains switched back to their initial polarizations while the *c* domains did not respond to the bias field (−3.3 V). The *c* domains were immobile during the electric loading, representing a frozen state. After one electric cycle, the entire domain configuration recovered to its original state.

Ferroelectric hysteresis loops of switchable area of domains vs. bias voltage directly indicate polarization-electric field relationship, which is plotted to represent the ferroelectric property in thin TEM samples[42]. Ferroelectric hysteresis loops at different

cycles (Supplementary Fig. 6) show that the switchable area of domains decreased as the number of cycles increased. Besides, the growth rate of the unswitchable domains (Fig. 2f) was increasing, indicating a degradation of ferroelectric property. The frozen state of the c domains can sustain long times after the experiment, and remains even 18 months after the cyclic electric loading (Supplementary Fig. 7).

It is worth noting that most of the $a_2^+$ domains, in which there were no stress-induced nanodomains, experienced 180° switching, while the switching process of the $a_2^+$ domains with stress-induced nanodomains inside was complicated in that multi-switching steps occurred via the formation of the intermediate ferroelastic domains[29]. The small red hollow arrows in Fig. 4c indicate the intermediate ferroelastic domains in an $a_2^+$ domain. The zipper-shaped domain wall, which is denoted by the small white hollow arrow in Fig. 4c, was also formed during the switching process. These observations are consistent with the literature demonstrating the switching behavior of PMN-0.38PT both without and with constraint[29]. However, despite the switching behaviors of the $a_2^+$ domains during electric cycling, the c domains remain unchanged in the $a_2^+$ domains during the poling cycle.

## Discussion

Charge accumulation is commonly cited as the origin of ferroelectric degradation[7]. However, the route from charge accumulation to ferroelectric degradation remains unclear. Our results provide direct evidence of charge accumulation at the domain walls and subsequently these charges promote the formation of frozen domains, leading to ferroelectric degradation. This picture of the dynamic process bridges the knowledge gap of how the space charges or injected charges contribute to ferroelectric degradation. As this study was performed on specimens with nanoscale dimensions, the physics of ferroelectric degradation demonstrated here might only be applicable to thin-film ferroelectrics and nanodevices. In spite of this, our work is thought-provoking as ferroelectric degradation in ferroelectric memories[2], MEMS and nanoelectromechanical system devices[1] is still a major issue to be tackled with.

In this work, we illustrate that frozen domains initiated from the typical polarization transition zones-domain walls. More completely, any polarization transition zones, including phase boundaries, interfaces and surfaces of ferroelectrics, could be potential sites for the formation of frozen domains during cyclic electric loadings. This is because the phase boundaries, interfaces and surfaces of ferroelectrics are not only the preferred sites where extra charges accumulate but also polarization transition regions[22]. It can be corroborated by the formation of some frozen domains near the lower edge of the sample in our experiment (Fig. 2a) as the sample edge is a reconstructed surface where the polarizations are randomly distributed[43]. We can make a reasonable conjecture that multi-phase coexisted regions[44], film/substrate interfaces, ferroelectric/electrode interfaces, and grain boundaries[45] could be the regions where frozen domains frequently appear under cyclic electric loadings.

In summary, we investigated the ferroelectric domain evolution of PMN-0.38PT single crystals using in-situ biasing TEM. The observed domain switching behavior under cyclic in-plane electric biasing showed that ferroelectric degradation could be induced by the formation of frozen c domains. The c domains formed from the original domain walls as the number of electric cycles increased. STEM-DPC results indicate that charges accumulated at the domain walls with the increase of the number of cyclic electric loadings. These accumulated charges served as compensation charges to stabilize the c domains. The c domains did not reverse in response to in-plane cyclic electric loading,

leading to the ferroelectric degradation of the material. This finding illustrates a dynamic process of ferroelectric degradation caused by the formation of frozen domains.

## Methods

**In-situ biasing TEM.** The PMN-0.38PT single crystal was grown by a modified Bridgman method[27]. It was selected as a model material for this work due to its simple and stable structure. PMN-0.38PT lamellae with dimensions of 8 μm× 4 μm× 1 μm (length × width × thickness) were produced using mechanical grinding followed by focused ion-beam processing. The PMN-0.38PT lamellae were fixed at a gap length of 4 μm between the inner two electric contacts of MEMS-based nano-chips by platinum deposition. The lamellae were thinned down by focused ion-beam to a thickness of about 100 nm. In-situ biasing TEM experiments were conducted with a Lightning holder produced by DENSsolutions in a JEM-2100 electron microscope operated at 200 kV. A bipolar bias in a triangular waveform (peak voltage of 4.7 V and frequency of 1/30 Hz) was applied along the ±[100] crystallographic direction of the specimen.

**STEM-DPC.** When a focused electron beam passes through a ferroelectric sample, the local electrostatic potential field causes a phase shift of the electron beam, which changes the intensity distribution of the corresponding convergent beam electron diffraction (CBED) pattern. The differential of this phase shift is linearly related to the position of the center of mass of the CBED pattern[46]. For a TEM sample, the phase shift of the transfer function of the sample is approximately proportional to its projected electrostatic potential. Accordingly, the differential of the phase shift is proportional to the projected electrostatic field of the sample[47]. The STEM-DPC technique relates the differential of the phase shift and the electrostatic potential field of the sample[34]. In this work, STEM-DPC was performed in an aberration-corrected FEI-Themis Z STEM operated at 300 kV. The configuration of the detector segments is illustrated in Supplementary Fig. 8. The images were obtained with a semi-convergence angle of 17.9 mrad and collection angle range of 9–51 mrad. A beam current of 50 pA, a dwell time of 10 μs per pixel and an image size of 1024 × 1024 pixel were utilized. Because STEM-DPC imaging is very sensitive to crystallographic orientation, domain switching is not desired during the application of the cyclic electric field. STEM-DPC images were taken before and after the application of 100 cycles of electric loadings in another PMN-0.38PT sample. In order not to induce domain switching (Supplementary Fig. 9), a bipolar bias in a triangular waveform with a frequency of 1/8 Hz and a peak voltage of 3.0 V, which is smaller than the switching voltage in this sample, was applied along the ±[100] crystallographic direction of the specimen.

## Data availability

The authors declare that the data supporting the findings of this study are available within this paper and its supplementary information files.

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

## Acknowledgements

The authors acknowledge the facilities and the scientific and technical assistance of the Microscopy Australia node at the University of Sydney (Sydney Microscopy & Micro-analysis). The authors also acknowledge the financial support from the Australian Research Council Discovery Project DP190101155.

## Author contributions

Q.H., Z.C., and X.L. designed and directed the study. Q.H. conducted the in-situ TEM experiments and performed data analysis. Z.C. and M.J.C. assisted in the experiment. F.W. and H.L. provided the samples. Q.H., Z.C., M.J.C., S.Z., F.L, Y.L., S.P.R., Y-W. M., and X.L. wrote the paper. All authors discussed the results and contributed to the paper.

## Competing interests

The authors declare no competing interests.
