## [Peer Review File · Nature Communications]

REVIEWER COMMENTS

Reviewer #1 (Remarks to the Author):

Huang et al. report the ferroelectric degradation by a cyclic electric field. It is interesting that c domain switched after hundreds of cycles of electric fields. Since the ferroelectric degradation has been extensively explored in the previous work, authors focus on the underlying mechanism of this polarization reversal failure. They claim that the accumulated charges during the cycling at the domain wall cause the nucleation of c domain and thus leads to the degradation. My concern is that experimental data presented in this manuscript do not sufficiently support this claim.

Authors use STEM-DPC to demonstrate the charge variation before and during electric fields. However, DPC contrast or intensity depend on the complicated imaging conditions such as focus, sample tilt and even the beam probe (Sci. Rep. 5, 10040; doi: 10.1038/srep10040 (2015)). My questions as below:

1. How can authors get a precise imaging condition before and during the cycling as a slight change in the focus and tilt would lead to the difference in the intensity?

2. Can authors give a quantitative analysis of the electric field? It is necessary to know the magnitude of the electric field if charge accumulated.

3. For a_1/a_2 domain, the red and green contrast (Fig.3) is linked to the polarization direction. In this case, DPC gives the information of build-in electric field, while in the a_1/a_2 boundary, it is the electrostatic field that cause the contrast variation. So can authors distinguished these two field types from DPC?

4. In Fig.3, authors demonstrate the charge accumulation at domain walls which would lead to the growth of c domain near them. However, in Fig. 2, the c domains start to nucleate near the surface (the lower part of the sample). If it is the case, the surface effect seems to dominate the c domain growth rather the charge accumulation.

5. In Fig. 2, there are some new domains formed in a_2+ area, which is similar to the mechanical case reported in Ref 29. As the cycling fields applied, the inverse piezoelectric effects need to be considered, which may change the stress of the sample and cause the domain switching or intensity change in DPC.

6. Authors attribute the 'frozen' c domain to the compensation charges on the surface as shown in Fig. 4b. The c domain is formed to compensate the accumulation charges on the domain walls. Why it again be compensated by the charges on the surface?

7. How long does the degraded states sustain without the cycling fields?

Other Comments:

1. Authors should give the configuration of the segmented detectors (e.g. how many segmented detectors, sample orientation relationship with respect to the detector segments).

2. The schematic in Fig.1c seems confusing. The split spots can only be distinguished at the upper left and bottom right part. Is this the real case?

3. Some Reference should be correctly formatted, like Ref 3, Ref 11, and etc..

Reviewer #2 (Remarks to the Author):

In this manuscript, the authors described the charge accumulation at domain walls of ferroelectric PMN-0.38PT ceramic during cyclic electric loading, which initiated formation of the new domains with out-of-plane polarization. The new domains were non-sensitive to the external electric field, resulting the ferroelectric degradation of PMN-PT. Overall, the experiments were carefully conducted and the manuscript were very well organized. The design and concept of this work proposed by the authors have novelty. This is an interesting paper and can be considered to accept after some revisions as

below. The authors claimed the new c domain was frozen to the applied electric field and confirmed its frozen behavior by using a TEM tool, as shown in Fig. 4c. However, it would be suitable that the authors directly investigate the ferroelectric properties of PMN-0.38PT as the number of applied electric cycles increased via ferroelectric characterization systems.

Responses to reviewers' comments

We sincerely thank the reviewers for their constructive comments and valuable suggestions. We have addressed all the reviewers' comments/suggestions and revised our manuscript accordingly. We believe that the quality of our manuscript has been greatly improved. Please see below our point-by-point responses to the comments/suggestions and the corresponding revisions. The revisions are highlighted below and in the revised manuscript.

Responses to Reviewer #1

Huang et al. report the ferroelectric degradation by a cyclic electric field. It is interesting that c domain switched after hundreds of cycles of electric fields. Since the ferroelectric degradation has been extensively explored in the previous work, authors focus on the underlying mechanism of this polarization reversal failure. They claim that the accumulated charges during the cycling at the domain wall cause the nucleation of c domain and thus leads to the degradation. My concern is that experimental data presented in this manuscript do not sufficiently support this claim.

Authors use STEM-DPC to demonstrate the charge variation before and during electric fields. However, DPC contrast or intensity depend on the complicated imaging conditions such as focus, sample tilt and even the beam probe (Sci. Rep. 5, 10040; doi: 10.1038/srep10040 (2015)).

Response: We thank the Reviewer for the thoughtful comment. We agree with him/her that STEM-DPC is sensitive to the imaging conditions. We confirmed that our STEM system is so stable that the focus, sample tilt and beam probe conditions do not change during in-situ biasing observations. We have conducted extra experiments and provided additional data below to support our claim.

1. How can authors get a precise imaging condition before and during the cycling as a slight change in the focus and tilt would lead to the difference in the intensity?

Response: We would like to point out that Figs. 3a and 3b were taken before and after cyclic electric loading, respectively, not during cyclic electric loading. We provided additional experimental data in the revised manuscript to prove that the electric loading processes did not change the focus and tilt in our in-situ experiments.

We validated the stability of the system by comparing two atomic-resolution STEM-HAADF images, before and after cyclic electric loading, without adjusting the focus and sample tilt. This solid proof is provided as Supplementary Fig. 1 in the revised manuscript.

Supplementary Fig. 1 Atomic-resolution STEM-HAADF images of a sample area (**a**) before and (**b**) after 50 cycles of electric loadings without any adjustment of sample tilt and focus. It is clear that the cyclic electric loading did not change the STEM-HAADF image. Because atomic-resolution STEM-HAADF images are sensitive to sample focus and tilt, the almost identical images in **a** and **b** confirm that the system is stable and that the in-situ experiments did not change the focus and tilt.

To verify the reliability and sensitivity of the STEM-DPC images in terms of the sample focus, we took a set of STEM-DPC images using various defocus conditions while keeping other parameters unchanged. These data are provided as Supplementary Fig. 2 in the revised manuscript.

Supplementary Fig. 2 Electric field maps taken at defocus of 0 nm (a), 10.02 nm (b), 5.41 nm (c), -4.37 nm (d) and -9.66 nm (e). The direction and amplitude of local electric fields are represented by colors, see the color wheel at the bottom of a, and image intensity, respectively. Grayscale images f, g, h and i were obtained by subtracting the image a from the images b, c, d and e, respectively. Comparison of b, c, d and e to a shows no difference in the intensity, as shown in f, g, h and i, respectively, indicating that defocus condition does not change the electron field maps observed from STEM-DPC images within a certain defocus range.

Based on the above Supplementary Figs. 1 and 2, we have also added the following sentence in the main text (page 9):

“Our experimental results suggest that the repeated electric loading processes did not change the focus and tilt (Supplementary Fig. 1) and that defocus condition did not change the electron field maps observed from STEM-DPC images within a certain defocus range (Supplementary Fig. 2).”

To rule out the other possibilities that may cause the change in STEM-DPC intensity, e.g., the instability of the electron probe with time, we provided another series of STEM-DPC images before and after electric loadings as Supplementary Fig. 4 in the revised manuscript.

Supplementary Fig. 4 A series of STEM-DPC images with different time and different numbers of electric loading cycles. Electric field maps taken at 0 minute (**a**), 8 minutes later (**b**), after 1 cycle (**c**) and after 100 cycles (**d**) of cyclic electric loadings. The direction and amplitude of local electric fields are represented by colors, see the color wheel at the bottom of **a**, and image intensity, respectively. Grayscale images **e**, **f** and **g** were obtained by subtracting the image **a** from the images **b**, **c** and **d**, respectively. There is no contrast change between **a** and **b**, indicating the stability of STEM-DPC. Also, there is no contrast change between **a** and **c**, indicating little charge accumulation after only one cycle. The black hollow arrows in **g** mark an area with clear intensity variation, indicating charge accumulation there as a result of many cycles of electric loadings. It is worth noting that the domain walls possess the capability of gathering charges, but this capability varies for each domain wall depending on many factors including sample loading history. This can be seen from **g** that the upper right domain shows distinct change in intensity after cyclic electric loading, while the intensity change of the other two domain walls at the bottom left part is not as obvious.

Based on the above Supplementary Fig. 4, we have also added the following sentence in the main text (page 10):

“To further validate the stability and reliability of STEM-DPC imaging, we provide additional data in Supplementary Figs. 3 and 4.”

The above experimental data prove that the system is stable and no sample tilting, focus and electron probe change during the in-situ experiment.

2. Can authors give a quantitative analysis of the electric field? It is necessary to know the magnitude of the electric field if charge accumulated.

Response: Thanks for the valuable suggestion. The homogeneity of the electric field for the commercial TEM holder used in this work is as high as 99% (<https://denssolutions.com/products/lightning/>). Figure R1 is copied from Ref. 30 (Garza, Héctor Hugo Pérez, et al. "MEMS-based sample carriers for simultaneous heating and biasing

experiments: A platform for in-situ TEM analysis." *2017 19th International Conference on Solid-State Sensors, Actuators and Microsystems (TRANSDUCERS)*. IEEE, 2017) that confirms the extremely high homogeneity of the electric field between the electrodes of a MEMS-based chip used in the same type of holder.

[Redacted]

Fig. R1 Electric field distribution between the inner two electrodes of a MEMS-based chip when an electric field of 200 kV/cm is applied (copied from Ref. 30).

Therefore, the applied electric field can be accurately calculated in our work, whose peak value was $\pm 1.175 \text{ MV m}^{-1}$ with a bipolar triangle wave (as indicated in the main text). We rewrote the following sentences in the revised manuscript (page 6) to avoid any confusion.

“The electric field E applied to the lamella can be calculated using the equation: $E = \frac{V}{d}$, where V and d are the applied voltage and the distance between the electrodes, respectively. Since the electric field between the two electrodes is homogeneously distributed and the distance between the two electrodes is $4 \mu\text{m}$, the peak applied electric field is $\pm 1.175 \text{ MV m}^{-1}$.”

3. For a1/a2 domain, the red and green contrast (Fig.3) is linked to the polarization direction. In this case, DPC gives the information of build-in electric field, while in the a1/a2 boundary, it is the electrostatic field that cause the contrast variation. So can authors distinguished these two field types from DPC?

Response: It should be noted that the STEM-DPC experiments were conducted on another PMN-0.38PT sample where a peak voltage of 3.0 V did not cause domain switching (see Supplementary Fig. 8). Therefore, no polarization reversal occurred during the DPC experiments and the reason for the DPC contrast variation at domain walls could only be electrostatic field change, which was caused by the charge accumulation at domain walls.

4. In Fig.3, authors demonstrate the charge accumulation at domain walls which would lead to the growth of c domain near them. However, in Fig. 2, the c domains start to nucleate near the surface (the lower part of the sample). If it is the case, the surface effect seems to dominate the c domain growth rather the charge accumulation.

Response: We appreciate the Reviewer's valuable comment. The occurrence of c domains is a random phenomenon in terms of time. However, the position of the nucleation is always the domain boundaries. In our experiment, the farthest c domains are more than 500 nm away from the lower surface of the sample. At such distance, the surface effect has minor impact on the formation of c domains, and the domain walls should be the dominant factor for the formation of c domains in that region. However, the surface could assist the formation of c domains as charges also prefer to accumulate there. In this case, the formation of c domains at the extreme lower edge could be a combined effect of the surface and the domain walls.

The above discussion has been given on page 14 in the original manuscript: "More completely, any polarization transition zones, including phase boundaries, interfaces and surfaces of ferroelectrics, could be potential sites for the formation of new frozen domains during cyclic electric loadings. This is because the phase boundaries, interfaces and surfaces of ferroelectrics are not only the preferred sites where extra charges accumulate, but also polarization transition regions."

5. In Fig. 2, there are some new domains formed in a_2+ area, which is similar to the mechanical case reported in Ref 29. As the cycling fields applied, the inverse piezoelectric effects need to be considered, which may change the stress of the sample and cause the domain switching or intensity change in DPC.

Response: Thanks also for the thoughtful comment. The nucleation of c domains occurred at the domain walls, where there were no nanodomains at all. In addition, if it is the inverse piezoelectric effect that made the intensity change in DPC, the change should be there in both domains and domain walls, not just the domain walls. Hence, the intensity in DPC cannot be mainly attributed to inverse piezoelectric effects.

In Fig. 3 and Supplementary Fig. 4, the DPC images were taken from domains without these nanodomains. It is worth noting that the bias we applied for DPC imaging did not cause domain switching in that sample. To strengthen the point that no domain switching took part during the STEM-DPC imaging, we have provided Supplementary Fig. 8 in the revised manuscript.

Supplementary Fig. 8 A series of TEM images showing no domain switching occurred in the PMN-0.38PT sample used for STEM-DPC imaging under a cyclic electric loading with a peak voltage of 3.0 V. The blue dot in each image indicates the applied voltage.

The biasing condition used for STEM-DPC imaging was already in the original Methods part (page 16) “In order not to induce domain switching (Supplementary Fig. 8), a bipolar bias in a triangular waveform with a frequency of 1/8 Hz and a peak voltage of 3.0 V, which is smaller than the switching voltage in this sample, was applied along the $\pm[100]$ crystallographic direction of the specimen.”

6. Authors attribute the ‘frozen’ c domain to the compensation charges on the surface as shown in Fig. 4b. The c domain is formed to compensate the accumulation charges on the domain walls. Why it again be compensated by the charges on the surface?

Response: Thanks for the valuable question. Domain walls are not only polarization transition zones, but also regions where charges accumulate. New domain formation and charge accumulation could occur synchronously or asynchronously. As shown in Supplementary Fig. 4g, the capability of gathering charges for each domain wall varied. Therefore, the occurrence of the frozen domains is a random phenomenon. We cannot foresee when it happens. But once the frozen domains form, the accumulated charges will compensate and stabilize them.

7. How long does the degraded states sustain without the cycling fields?

Response: The degraded states can sustain long times after the experiment, even over 18 months as is shown in our extra experimental data below. The corresponding content has been added in the revised manuscript (Supplementary Fig. 6).

Supplementary Fig. 6 The frozen domains remained 0 day (a), 7 days (b) and 18 months (c) after the cyclic electric loading. The white arrows indicate the frozen domains.

Based on the above Supplementary Fig. 6, we have also added the following sentence in the main text (page 12):

“The frozen state of the c domains can sustain long times after the experiment, and remains even 18 months after the cyclic electric loading (Supplementary Fig. 6).”

8. Authors should give the configuration of the segmented detectors (e.g. how many segmented detectors, sample orientation relationship with respect to the detector segments).

Response: Thanks for the suggestion. The configuration of the segmented detectors is added as below in the revised manuscript (Supplementary Fig. 7).

Supplementary Fig. 7 A schematic diagram of the orientation relationship between the detector segments and the sample. The opposite detector segments A and C are aligned in the direction of the electric field. The opposite detector segments B and D are aligned in the direction perpendicular to the electric field. The yellow shadow denotes the bright-field disk.

Based on the above Supplementary Fig. 7, we have also added the following sentence in the Methods part (page 16):

“The configuration of the detector segments is illustrated in Supplementary Fig. 7.”

9. The schematic in Fig.1c seems confusing. The split spots can only be distinguished at the upper left and bottom right part. Is this the real case?

Response: Our apologies for the confusion. A magnified Fig. 1c is presented in Fig. R2, and as can be observed, the diffraction spot array indicated by the two black arrows does not split. To clarify this, we have provided an explanation in the original submission of the manuscript (page 5) “The electron diffraction pattern shows the spot splitting caused by the 90° polarization rotation of a_1 domains and a_2 domains. The yellow and purple spots are contributed by the yellow (a_1 domains) and purple (a_2 domains) lattices, respectively.”

Fig. R2. A magnified Fig. 1c.

It is the real case in the diffraction pattern. A full image of the real diffraction pattern (to make the spot split clearer, only a quadrant located in the lower right part of the diffraction pattern is presented in Fig. 1e) is presented in Fig. R3. Only the diffraction spot array indicated by the frame does not split.

Fig. R3 A full image of Fig. 1e.

10. Some Reference should be correctly formatted, like Ref 3, Ref 11, and etc..

Response: We have corrected the reference format and checked all references to follow the same correct format.

Responses to Reviewer #2

In this manuscript, the authors described the charge accumulation at domain walls of ferroelectric PMN-0.38PT ceramic during cyclic electric loading, which initiated formation of the new domains with out-of-plane polarization. The new domains were non-sensitive to the external electric field, resulting the ferroelectric degradation of PMN-PT. Overall, the experiments were carefully conducted and the manuscript were very well organized. The design and concept of this work proposed by the authors have novelty. This is an interesting paper and can be considered to accept after some revisions as below.

Response: We thank the Reviewer for the very positive comments.

The authors claimed the new c domain was frozen to the applied electric field and confirmed its frozen behavior by using a TEM tool, as shown in Fig. 4c. However, it would be suitable that the authors directly investigate the ferroelectric properties of PMN-0.38PT as the number of applied electric cycles increased via ferroelectric characterization systems.

Response: We thank the Reviewer for the valuable suggestion. The degradation mechanism observed in this current research only applies to thin ferroelectrics. Therefore, ferroelectric property measurement in a bulk PMN-0.38PT (the fatigue in bulk PMN-0.38PT occurs at much higher cycling number, >1000 cycles) does not represent the case in nano-scale samples. Due to the small-size of the material, it is difficult to directly measure the polarization-electric field loop. However, the switchable area of domains *versus* bias voltage is a direct indication of the polarization-electric field relationship, which is plotted to represent the ferroelectric property in thin TEM samples¹. We added the loops of a switched area of a projected domain *versus* bias voltage in Supplementary Fig. 5 as follows:

Supplementary Fig. 5 Ferroelectric hysteresis loops of an area of projected a_1^- domain *versus* bias at 1, 130, 200 and 280 cycles. The initial a_1^+ domain marked with yellow shadow in the inset is used for the graph. This graph records the area of the a_1^+ domain switching to a_1^-

domain as the applied bias changes at different cycles. All the loops are asymmetric. At the peak positive bias (4.7 V), the switched area decreases as the number of cycles increases, indicating a degradation of ferroelectric property. This is due to the increased number of the frozen domains during the cyclic electrical loading.

Based on the above Supplementary Fig. 5, we have also added the following sentences in the main text (page 12):

“Ferroelectric hysteresis loops of switchable area of domains *versus* bias voltage directly indicate the polarization-electric field relationship, which is plotted to represent the ferroelectric property in thin TEM samples. Ferroelectric hysteresis loops at different cycles (Supplementary Fig. 5) show that the switchable area of domains decreased as the number of cycles increased. Besides, the growth rate of the unswitchable domains (Fig. 2f) was increasing, indicating a degradation of ferroelectric property.”

Correction:

In the end, we would like to make a revision on Fig. 2f. In the correction, the trend of the curve does not change, while the vertical scale is revised based on the new calculation. The revised graph is shown below.

f

Reference

1. Gao P, *et al.* Revealing the role of defects in ferroelectric switching with atomic resolution. *Nat Commun* **2**, 591 (2011).

REVIEWER COMMENTS

Reviewer #1 (Remarks to the Author):

The manuscript is improved and some of the comments are addressed. However, I am still not convinced that they have truly demonstrated the accumulated charges by DPC. Some issues which were raised in the first version remain:

1) The authors use the subtraction of DPC images before and after cyclic electric fields. They declare their imaging condition (tilt) is stable by showing the HAADF-STEM images, which is not persuading since a little tilt would not cause any difference in HAADF but can contribute to the DPC. And in Supplementary Fig.4e, f, the contrast difference can also be seen near the domain wall. So even though the DPC show contrast difference in the domain wall area, it does not mean the charges accumulation.

2) In the last version, I have asked "Can the authors give a quantitative analysis of the electric field? It is necessary to know the magnitude of the electric field if charge accumulated." I mean the magnitude of electric fields caused by the charge accumulation or the density of the accumulated charges. I think it is necessary to provide such information.

3) If the charges accumulate, I think it can be reflected by the EELS spectrum. If the authors can give the proof by EELS data, it would be more convincing.

Reviewer #2 (Remarks to the Author):

The authors provided the appropriate response and additional results in response to referees; as a result, the revised manuscript was improved by thorough revision process. The reviewer thinks that the manuscript is acceptable to this journal without any change.

Responses to reviewers' comments

We genuinely thank the reviewers for their comments and further suggestions. We have conducted extra EELS experiments and revised our manuscript to address the last reviewer's new comments/suggestions. The revisions have further strengthened our manuscript. Please see below our point-by-point responses to the comments/suggestions and the corresponding revisions. The revisions are highlighted below and in the latest revised manuscript.

Responses to Reviewer #1

The manuscript is improved and some of the comments are addressed. However, I am still not convinced that they have truly demonstrated the accumulated charges by DPC. Some issues which were raised in the first version remain:

1) The authors use the subtraction of DPC images before and after cyclic electric fields. They declare their imaging condition (tilt) is stable by showing the HAADF-STEM images, which is not persuading since a little tilt would not cause any difference in HAADF but can contribute to the DPC. And in Supplementary Fig.4e, f, the contrast difference can also be seen near the domain wall. So even though the DPC show contrast difference in the domain wall area, it does not mean the charges accumulation. If the charges accumulate, I think it can be reflected by the EELS spectrum. If the authors can give the proof by EELS data, it would be more convincing.

Response: We thank the Reviewer for the valuable suggestion. We understand the Reviewer's concern for the sake of scientific rigor. Therefore, additional electron energy loss spectroscopy (EELS) data are provided in this updated revision.

Based on the plasma theory, the change of charge density results in the shift of plasma peak¹. The change of electron density near the surface can be determined from the calculated plasma frequencies from EELS before and after the electric field².

We added the following Supplementary Fig. 5 showing the EELS data at a domain wall before and after 100 cycles of electric loading in the revised manuscript:

Supplementary Fig. 5 EELS spectra at a domain wall before (blue) and after (red) 100 cycles of electric loading. The spectra were acquired using the dual-EELS mode, at 300 kV with a semi-convergence angle of 17.9 mrad, a collection angle range of 48 – 200 mrad and a beam current of 40 pA. The dual-EELS technique acquires both the zero-loss and the low-loss/core-loss signals, guaranteeing the accuracy of peak shift determination³. The energy resolution of the EELS is 1.0 eV, which was measured from the full width at half maximum of the zero-loss peak with an energy dispersion of 0.025 eV/channel. Considering that the energy lost to generate a plasmon is typically in the range 5-25 eV⁴ and that the EELS with a higher energy loss can minimize the influences from other factors², the energy loss between 20 eV to 25 eV is selected for analysis. The plasma peak shifted 1.2 eV after cyclic electric loading, indicating a change in charge density at the domain wall. This confirms our DPC results in the main text.

2) In the last version, I have asked “Can the authors give a quantitative analysis of the electric field? It is necessary to know the magnitude of the electric field if charge

accumulated.” I mean the magnitude of electric fields caused by the charge accumulation or the density of the accumulated charges. I think it is necessary to provide such information.

Response: Based on the literature², the change in charge density can be calculated according to the shift in plasma energy from EELS. We added the following calculation in Supplementary Information:

Change in charge density at domain walls before and after cyclic electric loadings

According to the plasma theory, change in charge density ΔN can be expressed as²

$$\Delta N = \frac{\epsilon_0 m}{e^2} (\omega_{after}^2 - \omega_{before}^2) \quad (1)$$

where ω_{before} and ω_{after} are plasma frequencies before and after cyclic electric loadings, respectively, ϵ_0 is the vacuum permittivity, m is the effective electron mass, and e is the elementary charge of an electron. Plasma frequency can be obtained by the equation⁴

$$\omega_p = \frac{E_p}{\hbar} \quad (2)$$

in which \hbar is the Dirac constant and E_p is plasma energy. Therefore, the change in charge density can be rewritten as

$$\Delta N = \frac{\epsilon_0 m}{e^2 \hbar^2} (E_{after}^2 - E_{before}^2) \quad (3)$$

where E_{before} and E_{after} are the plasma energy before and after cyclic electric loadings, respectively. The change in charge density at domain walls is estimated to be 3.76 electrons/nm³ when $E_{before} = 23.4$ eV and $E_{after} = 24.6$ eV.

The estimation of the electric field caused by the charge accumulation can be simplified as the electric field between a parallel-plate capacitor using the following equation⁵

$$E = \frac{Q}{A\epsilon} \quad (4)$$

where Q is the amount of charges, A is the area to enclose the charge on the plate and ϵ is the permittivity of PMN-0.38PT. Q can be further expressed as

$$Q = \Delta NeV \quad (5)$$

in which V is the volume of the charges. Therefore, the electric field caused by the charge accumulation can be written as

$$E = \frac{\Delta Ned}{\epsilon} \quad (6)$$

where d is the layer thickness of accumulated charges.

We would like to emphasize that the above calculation is based on a simplified model, where the actual permittivity, effective mass and the layer thickness of accumulated charges can be altered case by case during an experiment. It is also worth of notice that the permittivity is also affected significantly by the environment and sample conditions, causing the variation of the magnitude of the electric field up to one order. Also, it is extremely difficult to measure the exact value of the permittivity of a sample with nanometer size. Therefore, we only provide an estimated magnitude of electric field in the Supplementary Information (as shown below):

In this simplified calculation, we assume the charges accumulated at the very surface of the sample and a thickness of a unit cell is considered. The adopted relative permittivity of PMN-0.38PT is 734 according to the literature⁶. The estimated electric field is then 37 MV/m. However, the permittivity of ferroelectric materials is highly dependent on the environment and sample conditions, causing the variation of the magnitude of the electric field up to one order. This electric field caused by accumulated charges is strong enough to maintain the c domains (the necessary electric field for domain switching in PMN-0.38PT thin lamella in this work is ~0.9 MV/m).

Based on the above Supplementary Information, we have also added the following sentences in the main text:

“With extra data from Electron energy loss spectroscopy (EELS), a powerful technique to evaluate the electronic state of materials⁷, we confirm the charge accumulation at domain walls. EELS was used to measure the plasma peak shift² at domain walls before and after cyclic electric loadings (Supplementary Fig. 5). The energy shift is 1.2 eV, corresponding to the change in charge density of 3.76 electrons/nm³ at the domain walls (See the calculation in Supplementary Information).” (page 10)

“It is also indicated by the calculations that the electric field induced by the accumulated charges is strong enough to stabilize the *c* domains (See Supplementary Information).” (page 11)

Responses to Reviewer #2

The authors provided the appropriate response and additional results in response to referees; as a result, the revised manuscript was improved by thorough revision process. The reviewer thinks that the manuscript is acceptable to this journal without any change.

Response: We appreciate the Reviewer for his/her recognition of our work.

References

1. Pines D. Elementary Excitations in Solids, WA Benjamin Inc. *New York*, (1963).
2. Meng Q, Xu G, Xin H, Stach EA, Zhu Y, Su D. Quantification of charge transfer at the interfaces of oxide thin films. *J Phys Chem A* **123**, 4632-4637 (2019).
3. <https://eels.info/about/techniques>.
4. Williams DB, Carter CB. The transmission electron microscope. In: *Transmission electron microscopy*. Springer (1996).
5. Griffiths DJ, Inglefield C. Introduction to Electrodynamics, 4th editon.). Cambridge University Press (2017).

6. Sun E, Cao W. Relaxor-based ferroelectric single crystals: Growth, domain engineering, characterization and applications. *Prog Mater Sci* **65**, 124-210 (2014).
7. Wei J, *et al.* Direct measurement of electronic band structures at oxide grain boundaries. *Nano Lett* **20**, 2530-2536 (2020).

REVIEWERS' COMMENTS

Reviewer #1 (Remarks to the Author):

In the revised manuscript, authors performed additional EELS analysis to make more convincing for the charge accumulation. I suggest the current version can be accepted for publish.

Responses to reviewers' comments

We sincerely thank the reviewers for their time and efforts in assessing our manuscript.

Response to Reviewer #1

In the revised manuscript, authors performed additional EELS analysis to make more convincing for the charge accumulation. I suggest the current version can be accepted for publish.

Response: We are happy that the Reviewer thought we have addressed all his/her concerns and suggested the current version can be accepted for publish.